# Haemolysis Detection in MicroRNA-Seq from Clinical Plasma Samples

**DOI:** 10.3390/genes13071288

**Published:** 2022-07-21

**Authors:** Melanie D. Smith, Shalem Y. Leemaqz, Tanja Jankovic-Karasoulos, Dale McAninch, Dylan McCullough, James Breen, Claire T. Roberts, Katherine A. Pillman

**Affiliations:** 1Flinders Health and Medical Research Institute, Flinders University, Bedford Park, SA 5042, Australia; shalem.leemaqz@flinders.edu.au (S.Y.L.); tanja.jankovickarasoulos@flinders.edu.au (T.J.-K.); dylan.mccullough@flinders.edu.au (D.M.); claire.roberts@flinders.edu.au (C.T.R.); 2Adelaide Medical School, University of Adelaide, Adelaide, SA 5000, Australia; dale.mcaninch@gmail.com; 3Indigenous Genomics, Telethon Kids Institute, Adelaide, SA 5000, Australia; jimmy.breen@telethonkids.org.au; 4College of Health & Medicine, Australian National University, Canberra, ACT 2601, Australia; 5Centre for Cancer Biology, University of South Australia/SA Pathology, Adelaide, SA 5000, Australia

**Keywords:** microRNA, plasma, biomarker, prediction, haemolysis, bioinformatics

## Abstract

The abundance of cell-free microRNA (miRNA) has been measured in blood plasma and proposed as a source of novel, minimally invasive biomarkers for several diseases. Despite improvements in quantification methods, there is no consensus regarding how haemolysis affects plasma miRNA content. We propose a method for haemolysis detection in miRNA high-throughput sequencing (HTS) data from libraries prepared using human plasma. To establish a miRNA haemolysis signature we tested differential miRNA abundance between plasma samples with known haemolysis status. Using these miRNAs with statistically significant higher abundance in our haemolysed group, we further refined the set to reveal high-confidence haemolysis association. Given our specific context, i.e., women of reproductive age, we also tested for significant differences between pregnant and non-pregnant groups. We report a novel 20-miRNA signature used to identify the presence of haemolysis in silico in HTS miRNA-sequencing data. Further, we validated the signature set using firstly an all-male cohort (prostate cancer) and secondly a mixed male and female cohort (radiographic knee osteoarthritis). Conclusion: Given the potential for haemolysis contamination, we recommend that assays for haemolysis detection become standard pre-analytical practice and provide here a simple method for haemolysis detection.

## 1. Introduction

MicroRNAs (miRNAs) represent a class of short, ~22-nucleotide single-stranded non-coding RNA transcripts found in the cytoplasm of most cells that act directly as post-transcriptional regulators of gene expression [1,2] and also coordinate extensive indirect transcriptional responses [3]. In their canonical action, miRNAs mediate the expression of specific messenger RNA (mRNA) targets by binding to the 3′-untranslated region (UTR) of transcripts by either repressing translation or marking them for degradation [4]. In the canonical miRNA pathway, target specificity requires exact nucleotide-sequence complementarity between the miRNA ‘seed’ region (the first 2–7 bases at the 5′ end of the mature miRNA transcript) and the 3′-UTR of the mRNA. Importantly, miRNAs demonstrate tissue, temporal, and spatial expression specificity and are known regulators of development, with most mammalian mRNAs harbouring conserved targets of one or many miRNAs [1,2,5].

miRNA expression is both temporally and spatially tissue-specific, with transcripts identified beyond the cells in which they were synthesized, in various body fluids, including urine, saliva, and blood plasma [6]. Circulating cell-free miRNAs identified in plasma are packaged in microvesicles such as exosomes [7,8] or bound to protein complexes, such as argonaute 2 (Ago2), nucleophosmin 1 (NPM1), and high-density lipoprotein (HDL) [9,10,11], making them exceptionally stable [6]. This stability, coupled with their minimally invasive accessibility, has suggested circulating cell-free miRNAs as an important resource for the identification of novel biomarkers.

Whilst much progress has been made in the search for novel miRNA biomarkers of disease processes [12,13,14], outcomes of this research approach are often inconsistent or even contradictory [6]. There are many reasons for this, including variations in enrichment, extraction, and quantification methods, variation among individuals, lack of consensus regarding optimal reference miRNA for normalization, and the difficulty in quantifying both the amount and quality of RNA transcripts from blood plasma samples [15,16]. An important but often overlooked factor is the potential for sample haemolysis during blood collection or sample preparation resulting in miRNA from lysed red blood cells (RBCs) being spilled into and retained within the plasma sample to be assayed [15].

The issue of haemolysis altering the miRNA content of plasma and the potential for confounding biomarker discovery has been reported previously [15,17,18]. Using RT-qPCR, Kirschner and colleagues [15] showed that contamination of plasma samples with the miRNA content of RBCs changed the abundance of both miR-16 and miR-451a. This, in turn, altered the relative abundance of potential biomarkers for mesothelioma and coronary artery disease, including miR-92a and miR-15. Using the same technique, Pritchard et al. [18] demonstrated in plasma that 46 of the 79 circulating miRNA cancer biomarkers were highly expressed in more than one blood-cell type, noting that the effects of sample-specific blood-cell counts and haemolysis can alter miRNA biomarker levels in a single patient sample up to 50-fold. As a result, the authors emphasized caution in classifying blood-cell-associated miRNAs as biomarkers, given the possible alternate interpretation.

Haemolysis is associated with either blood collection or RNA extraction and sample preparation. Thus, despite differences between the quantification methods, high-throughput sequencing data used in our study is equally susceptible to the confounding effects of sample haemolysis on miRNA abundance levels in plasma as RT-qPCR. There are currently two commonly used gold standard approaches in the assessment of haemolysis in plasma: (1) delta quantification cycle (ΔCq), where expression levels of a known blood cell-associated miRNA (miR-451a) and a control miRNA (miR-23a-3p) are determined based on the difference between the two raw Cq values; and (2) spectrophotometry, where absorbance is measured at 414 nm (A414) with the use of a spectrophotometer. In the case of ΔCq assessment, miR-451a is known to vary and miR-23a-3p is known to be invariant in plasma affected by haemolysis [15,16]. Using spectrophotometry, haemolysis is quantified by assessing the presence of cell free haemoglobin by measuring the absorbance at 414 nm, the absorbance maximum of free haemoglobin [19,20]. In controlled experiments, haemolysis is highly correlated with raw A414. Patient samples, however, may sometimes be affected by sample interferences, such as lipaemia. In these cases, more sophisticated absorbance-based corrections to the raw A414 can improve the accuracy of haemolysis detection [21]. A third, less commonly used, method—ELISA—can also be used to detect haemolysis [22]. All three methods require access to sufficient amounts of the original plasma sample and the laboratory equipment required to perform the assays. Free access to a web tool that can perform in silico assessment of RBC contamination in human plasma would be of exceptional value to the research community.

Whilst it is well established that haemolysis frequently occurs during extraction or processing of blood samples, the assessment of RBC contamination is rarely mentioned in publications. It is even rarer that the results of any such testing are present in the metadata assigned to publicly available sequencing data. There is currently no publicly available tool for analysis of haemolysis without access to the physical plasma specimen. Although the theory underlying identification of haemolysis in plasma is relatively straightforward, surprisingly, to our knowledge, this has never before been extrapolated into a data-only in silico approach. The paucity of haemolysis information in the context of publicly available datasets combined with the lack of tools to identify affected datasets after the fact substantially limits the utility of this data resource and reproducibility of research findings. Further, it increases the risk that results obtained may unwittingly represent blood cell-based phenomena, rather than signatures of the pathology of interest.

In this study, we assessed miRNA abundance in HTS data from libraries prepared using human plasma from pregnant and non-pregnant women of reproductive age. Using a set of samples with confirmed haemolysis (ΔCq (miR-23a-3p-miR-451a), we established a set of 20 miRNAs differentially abundant between plasma from samples with and without substantiated haemolysis. Using the expression values of these 20 miRNAs as a ‘signature’ of haemolysis, we calculated the difference between the mean normalized expression levels of these miRNAs compared to those of all other miRNAs (as a ‘background’ set). This produced a quantitative metric representing the strength of the evidence of haemolysis in an individual sample. When this metric is interpreted in the context of other samples, it can be used to identify sample(s) that display substantial evidence of haemolysis. Where direct assessment of haemolysis cannot (or has not) be undertaken using one of the aforementioned assays, our method allows the researcher to consider discarding problematic sample data from further analyses or using caution in their interpretation. We consulted the EMBL-EBI Expression Atlas (ebi.ac.uk) to ensure all signature miRNAs were identified in multiple human tissue types (male and female) and had no known developmental stage association. For ease of application, we developed this method into a web-based Shiny/R application, DraculR (a tool that allows a user to upload and assess haemolysis in high-throughput plasma miRNA-seq data), for use by the research community [23].

## 2. Materials and Methods

### 2.1. Patient Information

Our analysis involved two prospectively collected cohorts: the first a cohort of women of reproductive age recruited at the Adelaide Medical School during 2005–2006 who were not pregnant at the time of blood collection, and the second a cohort of women undergoing elective termination of pregnancy at the Pregnancy Advisory Centre, Queen Elizabeth Hospital during 2016–2019. To test for differences among the means of our patient characteristics, we performed Student’s t-tests on dCq (miR-451a - miR-23a-3p) within both the pregnant and non-pregnant groups and across the cohort as a whole for each characteristic, splitting age, BMI, and gestational age into equal-sized groups. For each test the *p*-value was > 0.29, indicating no significant differences. All patient samples were collected with written, informed consent. Patient characteristics are shown in Table 1, with more detail regarding ethnicity and pregnancy status in Appendix A.

### 2.2. Sample Collection

Peripheral blood (9 mL) was collected with informed, written consent from women undergoing elective terminations of otherwise healthy pregnancies. Blood was collected into standard EDTA blood tubes pre-termination and stored on ice until processed. Whole blood underwent centrifugation at 800 × *g* for 15 min at 4 °C before plasma removal and then spun for a further 15 min to ensure any remaining cellular debris, including cell membranes from lysed red blood cells, was removed. All samples were stored at −80 °C until further processing. Termination samples were collected from the Pregnancy Advisory Centre (PAC), Woodville, South Australia. Blood was also collected with informed, written consent from non-pregnant volunteers at the Adelaide Medical School. Following collection, blood tubes were stored on ice until processing. Whole blood underwent centrifugation at 1015 × *g* for 10 min at 4 °C. Approximately 4–6 mL plasma was collected in 2 mL aliquots. 500 μL plasma (the supernatant) were aliquoted into clean tubes, and the pellet containing any remaining blood cells at the bottom of the tube was discarded. All samples were stored at −80 °C until further processing.

### 2.3. RNA Extraction and Library Preparation

For publicly available data previously published by us, raw sequencing reads were downloaded from NCBI GEO, Study GSE151362 [24] and combined with new small-RNA dataset with sequencing as follows. miRNA was isolated from 200 μL plasma samples using the Qiagen miRNA serum/plasma kit (Qiagen, Hilden, Germany) according to the manufacturer’s instructions and stored at −80 °C.

Library preparation and sequencing was performed using Qiagen (Valencia, CA) with the QIAseq miRNA Library and QIAseq miRNA 48 Index IL kits as per the manufacturer’s instructions (Qiagen, Hilden, Germany). Amplified cDNA libraries underwent single-end sequencing by synthesis (Illumina v1.9) using the Illumina NovaSeq 75 bp single-end read sequencing on miRNA libraries from 42 plasma samples taken from 10 non-pregnant and 32 pregnant women aged 16 to 44 years.

### 2.4. Haemolysis Detection by RT-qPCR

Plasma samples were examined with Qiagen (Hilden, Germany) for haemolysis based on the expression levels of two miRNAs: miR-451a and miR-23a-3p. miR-451a (previously named miR-451) is known to be highly expressed in red blood cells, whereas miR-23a-3p is known to maintain stable abundance levels in plasma. After RNA extraction and cDNA synthesis, 2 μL of RNA was reverse-transcribed in 10 μL reactions using a miRCURY LNA RT kit (Qiagen version 5). Each RT was performed using an artificial RNA spike-in (UniSp6). cDNA was diluted 50× and assayed in 10 μL PCR reactions according to the protocol for miRCURY LNA miRNA PCR, and each miRNA was assayed once by PCR using the assays for miR-23a-3p and miR-451a. In addition to these miRNA assays, the RNA spike-ins were assayed. The amplification was performed in a LightCycler 480 real-time PCR system (Roche, Sydney, Australia) in 384-well plates. The amplification curves were analyzed using in-house software, both for determination of Cq (by the 2nd derivative method) and for melting curve analysis. The raw data were extracted from the LightCycler 480 software. The evaluation of expression levels was performed based on raw Cq values. According to the Qiagen protocol for haemolysis detection using the ΔΔCq method, samples with ΔCq <7 for these two miRNAs were considered clear of contamination, ΔCq >7 was considered contaminated, and ΔCq = 7 was considered borderline.

### 2.5. miRNA Annotation and Abundance

Read quality-control metrics were assessed using FastQC [25] (http://www.bioinformatics.babraham.ac.uk/projects/fastqc/) to check for per base sequence quality, sequence-length distribution, and duplication levels. Adapter detection and trimming were performed using Atropos [26]. Alignment was performed using BWA version 0.7.17-r1188 (GRCh38) [27]. UMI-tools was used to collapse duplicate reads mapped to the same genomic location with the same UMI barcode. Quality-control metrics were reported using multiQC [28]. Read counts for mature miRNAs were determined using an in-house script [29] with miRNA annotation from miRBase version 22.0 [30,31] (http://www.mirbase.org). miRNA counts were quantified as counts per million (CPM) miRNA reads.

### 2.6. Analysis of Potential Confounding Factors

All profile and expression analyses were conducted in the R statistical environment (v.4.0.2), using the edgeR (v.3.16.5) [32] and limma (v.3.30.11) [33] R/Bioconductor packages. Prior to conducting the differential expression analysis between haemolysed and non-haemolysed expression data, we considered the effect of participant characteristics, such as sex, age, smoking, pregnancy status, and ethnicity. Sex was not included here, as all samples were taken from female participants. Maternal age was excluded from the final regression model, as there was no strong evidence of association with the outcome, and hence, considering the sample size, a simpler model was chosen to preserve degrees of freedom. Differential miRNA abundance between pregnant and non-pregnant groups was tested using only samples with ΔCq (miR-23a-3p-miR-451a) < 7 (i.e., not haemolysed). miRNA identified as differentially abundant between samples from pregnant and non-pregnant women was removed from the final set of haemolysis signature miRNA when calculating the haemolysis metric for our pregnant/non-pregnant cohorts. There were three independent sequencing batches in the data analyzed here, which are detailed in Appendix A. As such, sequencing batch was also included in all regression models.

### 2.7. Identification of Haemolysis miRNA Signature

Prior to defining a collection of haemolysis informative miRNAs for the 121 samples analyzed here, pre-filtering steps were undertaken: 1) mature miRNA with fewer than five reads was reduced to zero independently for each sample, and 2) miRNA with fewer than 40 counts per million (CPM) in the haemolysed group (*n* = 12) was removed from further consideration. This was done to ensure only highly abundant miRNA likely to be present in most samples remained. The trimmed means of M values (TMM) normalization method was used to correct for differences in the underlying distribution of miRNA expression [34]. Next, we used limma [33] to obtain the fold change of each miRNA between the haemolysed (*n* = 109) and non-haemolysed (*n* = 12) groups to identify miRNAs that were more abundant in the plasma affected by haemolysis. To ensure the haemolysis miRNA signature was robust, we took the intersection of the 60 miRNAs from each category of highest expression and lowest adjusted *p*-value and miRNAs with a log_2_FC > 0.9, revealing a set of 20 high-confidence miRNAs. To further refine the set of haemolysis informative miRNAs, we used limma to calculate the fold change for each miRNA between the samples from pregnant and non-pregnant women not affected by haemolysis and removed any of the high-confidence miRNA which was also differentially abundant in pregnancy. The workflow, source code, and input files associated with this research are available at (https://github.com/mxhp75/haemolysis_maternaPlasma.git).

### 2.8. Classification—Haemolysis Metric

To classify the data coming from samples as haemolysed, borderline or unaffected, we first focused on samples from the non-pregnant group. For these, we subset the miRNA read count table into miRNA from the high-confidence haemolysis informative miRNA (*n* = 20) and all others (*n* = 169). Using this data partition, we calculated the geometric mean of the distribution of read counts using the psych package (v1.8.12) [35] and subtracted the geometric mean of the counts of ‘other’ miRNA from that of the ‘haemolysis informative’ miRNA. Next, for samples from the pregnant group, we performed the same calculations described above after first discarding miRNA that was associated with pregnancy.

### 2.9. Data Availability

The dataset(s) supporting the conclusions of this article are available in: NCBI’s Short Read Archive (SRA) [36] and through BioProject accession number PRJNA824637 (https://www.ncbi.nlm.nih.gov/sra/PRJNA824637); and for previously published data, BioProject accession number PRJNA635621.

## 3. Results

### 3.1. High-Throughput Sequencing

Using libraries with > 1 million reads for analyses, we obtained 121 libraries with an average of ~3.49 million reads per sample (range ~1.00–18.64 million reads). RT-qPCR was used to analyze ΔCq (miR-23a-3p-miR-451a), where the ratio of miR-23a-3p to miR-451a (or ΔCq (miR-23a-3p-miR-451a) ≥ 7) correlated with the degree of haemolysis. We identified 12 plasma samples with a ΔCq of 7 or above (Appendix A) and determined that there was no difference in the proportion of haemolysed and non-haemolysed data in the exclusion of samples, due to low library size (Fisher’s exact test *p*-value = 0.7). Sequence alignment was performed using BWA [27] to the human genome (version GRCh38) and miRNA read counts were generated by mapping to human miRBase v22 [30,31] identifying 1133 mature miRNAs. Further analysis of non-miRNA small RNA species identified antisense, intergenic, and intronic (average of 13.4%, 16.9%, and 8.6% respectively) transcripts, as well as a small abundance of long non-coding and protein-coding RNA, rRNA, snRNA, and snoRNA transcripts (Appendix A).

To analyse the effects of haemolysis on miRNA-expression data from next-generation sequencing, we first determined the number of unique mature miRNAs identified in each of our samples and analyzed the data relative to read depth. Using an analysis of variance (ANOVA) we identified a significant difference between the haemolysed and non-haemolysed samples (*p* < 0.05), with haemolysis being frequently associated with fewer mature miRNA species detected at a given read depth (Figure 1).

### 3.2. MicroRNA Haemolysis Signature Set

To ensure that miRNAs identified here were representative of those found in a broad set of plasma samples, we first filtered to discard miRNAs of low abundance. After filtering, 189 highly abundant miRNA remained (Appendix A). Differential expression analysis comparing miRNA read counts identified 100 miRNAs with a higher abundance in haemolysed compared to non-haemolysed samples (statistically significant differentially expressed miRNA, false discovery rate (FDR) < 0.05, with a log_2_ fold change (log_2_FC) > 0) (Appendix A). We further ranked the differentially expressed miRNAs based on log_2_FC, FDR and abundance levels, and subset the list such that only miRNAs that had a log_2_FC > 0.9 and were in the top 60 percent of each of the FDR and abundance rank criteria remained. This resulted in a high-confidence set of 20 miRNAs indicative of a haemolysis signature (Table 2).

For in silico assay of haemolysis in our data, we further removed miRNAs that were differentially abundant between the PAC (pregnant) and NPC (non-pregnant) cohorts to avoid confounding miRNA associated with haemolysis with those associated with pregnancy. Differential expression analysis of miRNA read counts from pregnancy and non-pregnancy samples identified 127 miRNAs (FDR < 0.05) that were significantly differentially expressed between the groups (Appendix A). Strikingly, one of our first observations highlighted the importance of including haemolysis analysis as an adjunct in our study: miR-451a, which is the sole haemolysis signature miRNA used in the current ΔCq (miR-23a-3p-miR-451a) gold standard method for haemolysis detection, was discovered to be highly correlated with pregnancy status, indicating a strong confounding factor in pregnancy studies when haemolysis levels are estimated using RT-qPCR alone. Accordingly, when analysing the data from our pregnancy cohort, miR-451a was removed from calculations, along with nine other miRNAs that were differentially expressed between the pregnant and non-pregnant groups from the core set of haemolysis signature miRNA. This resulted in 10 miRNAs remaining for evaluation of haemolysis levels. Note that removing miRNAs from the signature set using our method also excluded those miRNA from the calculation of the distribution of background miRNAs.

Incorporating concepts from previous RT-qPCR analyses of haemolysis, we established a new measure of the inclusion of RBC-associated miRNA in human plasma. After establishing the 20-miRNA signature associated with RBC content inclusion, we determined the geometric mean of the distribution of miRNA read counts as an appropriate measure of abundance and summary statistic. Using this summary statistic, our method calculated a ‘haemolysis metric’, defined as the difference between the geometric means of the normalized abundance levels of the haemolysis miRNA signature set compared to that of all other miRNAs (the ‘background’ set). Note that in a case–control study, to reduce the risk of confounding the haemolysis metric with experimental variables, any miRNA known to be differentially expressed between groups should be excluded from both the signature, and background sets should be reduced to exclude any miRNA known to be differentially expressed between groups. In this case, the geometric mean of the reduced signature set will be calculated, as defined in Equation (1). Let $Z_{x}$ be the miRNA reduced signature set (log_2_ CPM counts) and $Z_{y}$ be the background miRNA set (log_2_ CPM counts), where $x=1,2,3,\ldots ,p_{1}$ with $p_{1}=$ the number of miRNAs in the reduced signature set and $y=1,2.3\ldots ,p_{2}$ where $p_{2}=$ the number of miRNAs in the background and i= 1,2.3,…,n  where n= the sample size after filtering:
(1)$$Haemolysis\;Metric_{i}=\sqrt[{p_{1}}]{\overset{p_{1}}{\underset{x=1}{\prod }}Z_{x_{i}}}-\sqrt[{pP_{2}}]{\overset{p_{2}}{\underset{y=1}{\prod }}Z_{y_{i}}}$$

Prior to establishing a threshold for the new haemolysis metric, we measured the linear dependence between the new haemolysis metric and the ΔCq (miR-23a-3p-miR-451a) metric by calculating Pearson’s correlation. Our results indicated a Pearson’s correlation coefficient of 0.64 (*p* < 0.0001) (Figure 2a) and also examined the linear relationship between the ΔCq (miR-23a-3p-miR-451a) metric and each individual signature set miRNA (Appendix A). With confidence in the correlation, to establish a threshold for the haemolysis metric, we compared the results of the ΔCq (miR-23a-3p-miR-451a) and summary statistic methods directly. Briefly, we compared the haemolysis metric to the ΔCq (miR-23a-3p-miR-451a) results for sample and established a cut-off criterion for inclusion into the Clear (no haemolysis detected) and Caution (haemolysis detected) groups (Figure 2a). We chose a threshold of ≥ 1.9 for the assignment of ‘Caution’ to individual samples based on the minimum summary statistic difference of samples assayed using the ΔCq (miR-23a-3p-miR-451a) metric of ≥ 7 (Figure 2a) and the minimal overlap between the distribution of the haemolysis metric in haemolysed compared to non-haemolysed samples (Figure 2b). Where a sample is assigned ‘Caution’, researchers are advised to consider removing the sample or to continue with caution. Given the correlation of the two metrics is imperfect and the arbitrary nature of choosing any cut-off, samples with a haemolysis metric close to the 1.9 cut-off may be interrogated further prior to any decision to retain or remove. Of the 121 samples assayed, 25 samples met the criteria for Caution. Of these, 12 were previously determined as haemolysed or borderline using the ΔCq (miR-23a-3p-miR-451a) assay. We found that all samples identified as ΔCq ≥ 7 (Figure 2a, scarlet) were above the criteria for the haemolysis metric (Figure 2a, horizontal grey bar; threshold ≥ 1.9). Further, we identified 13 samples with a haemolysis metric ≥ 1.9 not included in the ΔCq (miR-23a-3p-miR-451a) criteria.

To further validate the consistency and accuracy of the miRNA signature set, we performed two additional analyses. Firstly, we compared the results of the two gold standard methods of determining haemolysis: ΔCq (miR-23a-3p-miR-451a) and spectrophotometry absorbance A414 and the haemolysis metric described here (Appendix A). These results showed a high correlation (Pearson’s R > 0.82) among all methods, with the correlations between the haemolysis metric and the two other methods (R = 0.87 for A414, R = 0.90 for ΔCq (miR-23a-3p-miR-451a)) being higher than that between the two gold standard methods (R = 0.82). This is consistent with the haemolysis metric being an accurate and useful marker of haemolysis.

Next, using public plasma datasets for a) prostate cancer [37] and b) early radiographic knee osteoarthritis [12], we demonstrated application of the haemolysis metric to these case studies, including the exclusion of miRNAs suspected or confirmed to be differentially abundant for reasons unrelated to haemolysis (osteoarthritis: Appendix A; prostate cancer: Appendix A). First, we calculated a sequencing-based proxy of the ΔCq (miR-23a-3p-miR-451a) haemolysis metric (by subtracting the log_2_ (CPM) values for miR-451a from miR-23a-3p) and demonstrated that this metric was strongly correlated (R = 0.82) with ΔCq (miR-23a-3p-miR-451a) (Appendix A). We then validated both a) the reproducibility of the relationship between the abundance of individual signature miRNAs and the proxy miR-451a:miR-23a-3p metric, and b) the consistency between the sequencing-based proxy of ΔCq (miR-23a-3p-miR-451a) and the calculated haemolysis metric. Importantly, these datasets contained samples from both males and females (osteoarthritis) or male individuals only (prostate cancer), thus validating the approach beyond female individuals. We observed a consistent correlation among the many miRNAs in the signature set with the proxy miR-451a:miR-23a-3p metrics and between the proxy miR-451a:miR-23a-3p metric and the haemolysis metric, and no substantial difference in the relationships for male and female individuals (Appendix A). For each dataset, a small number of miRNAs (miR-191-5p for osteoarthritis [38], miR-30c-5p and miR-191-5p for prostate cancer [37]) had evidence or hints in the literature of a role in the biology of the dataset. In accordance with our recommendations, these were excluded from calculations of the haemolysis metric for the relevant dataset. This choice was supported by the anticipated marked lack of positive correlation between the expression of these miRNAs and the proxy miR-451a:miR-23a-3p metric (Appendix A). These examples reinforce the need for careful consideration of confounding signatures when calculating the haemolysis metric. In cases such as these, we recommend excluding from both the signature set and background any miRNAs known or suspected to relate to the study biology before calculating the haemolysis metric.

## 4. Discussion

Through an analysis of differential miRNA expression in samples whose haemolysis levels were known, we identified a novel 20 miRNA signature indicative of haemolysis. Given our hypothesis that plasma samples contaminated with RBC content would contain proportionally higher levels of many RBC-associated miRNAs, not just miR-451a, we established a method using a group of background miRNAs as a reference. Accordingly, as a group, signature miRNAs (miRNAs abundant in red blood cells) were shown to be more highly abundant in samples contaminated with RBCs. The degree of this change can be used as a measure of RBC content contamination and quantified by comparing the geometric means of the expressions of RBC signature miRNAs to that of the background set of miRNAs. We further established that where a comparison between conditions is considered, e.g., in a biomarker-discovery experiment, any miRNA known to be associated with the condition for which the biomarker is proposed should be removed to prevent confounding between the condition of interest and the quantification of RBC-associated miRNA inclusion.

Our experimental results demonstrate that it is possible to identify a haemolysis signature in silico, avoiding the effort and expense of lab validation, and in situations where blood plasma samples are exhausted, otherwise unavailable or cost-prohibitive to assay using current gold standard approaches. Given the limited access to physical samples associated with publicly available data, the haemolysis metric technique introduced here provides the research community with an alternative method for haemolysis detection. Our method also formed the basis for the development of a publicly available tool (DraculR: A web-based application for in silico haemolysis detection in high throughput small RNA sequencing data).

Among the haemolysis miRNA signatures was miR-451a (previously named miR-451), commonly associated with RBC contamination and used in the calculation of ΔCq (miR-23a-3p-miR-451a). However, we removed miR-451a together with nine other miRNAs from our calculation of distribution difference, due to changes in miRNA abundance associated with pregnancy. During pregnancy, total blood volume increases, varying between 20% to 100% above pre-pregnancy levels. This change, however, is not uniform across all blood components, as plasma volume increases proportionally more than the RBC mass [39]. This is an important consideration, and highlights the limitation of the current gold standard approach that uses two miRNAs, rather than a larger signature set, to calculate a measure of contamination. If, as in this example, the abundance of either miRNA used to determine the ΔCq is also affected by the condition or pathology under investigation, the issue is twofold. Firstly, you may identify miR-451a as being differentially abundant in the pathology of interest and propose its use as a biomarker, only to find that it is confounded by haemolysis. Secondly, you may, using the ΔCq calculation, classify samples as haemolysed when the change in miR-451a abundance is more appropriately associated with the pathology of interest. By establishing a larger signature set of miRNAs to detect haemolysis in small-RNA sequencing from human plasma, we hope to provide a resource to the community. To overcome issues identified in previous studies, the flexibility and redundancy included in our metric buffer against the issue of confounding conditions of interest with the measure of haemolysis.

We found a wide range in the overlap between the miRNAs identified as useful for the detection of haemolysis and those previously reported as markers of haemolysis contamination. Shkurnikov and colleagues used microarray analyses to correlate miRNA expression with haemoglobin concentration evaluated using spectrophotometry [40]. Their results were highly consistent with ours: of the nine miRNAs they reported that were significantly correlated with haemoglobin concentration, three were included in our signature set (miR-451a, miR-17-5p, and miR-20b-5p) and five others showed concordant differential abundance in our dataset (miR-486-5p, hsa-miR-16-5p, hsa-miR-93-5p, hsa-miR-20a-5p, and hsa-miR-107), though they were not chosen for the signature set, due to our stringent selection criteria including minimum miRNA abundance. Another microarray study [41] had similar consistencies and differences, with all five key haemolysis marker miRNAs reported being differentially abundant in our data (miR-486-5p, miR-92a-3p, miR-16-5p, and miR-22-3p), but only one (miR-451a) being in the signature set.

Several other studies have found limited overlap [15,18,42]. However, the differences in study methodology, as well as research question, are likely to contribute to these differences. Importantly, the method of quantification of miRNAs in these studies was targeted RT-qPCR, rather than transcriptome-scale approaches, such as HTS (used here) or microarrays [40]. The limitations of RT-qPCR to investigate which miRNAs are affected by haemolysis has been identified previously [43]. Given our use of HTS, our experiment was able to identify differential abundance in miRNAs not quantified in Kirschner et al. [15], Pritchard et al. [18], or McDonald et al. [42], meaning that whilst many of the miRNAs associated with previous haemolysis work were also associated here, including miR-16-5p, miR-486-5p, and miR-92a-3p, many of these were not included in the final miRNA haemolysis metric signature set, as they failed to pass filtering criteria for log_2_FC and expression level. Secondary to the technical differences introduced by using different miRNA quantification technologies, it is important to note that all plasma samples used here to establish which miRNAs were affected by haemolysis in our primary dataset were taken from adult women of reproductive age. No sex or age information was included with either of the compared studies, although it is likely these samples included specimens from men and women. To confirm that the use of the miRNA signature set to evaluate haemolysis could be generalized to male individuals, we performed two additional validation analyses to evaluate the relationship between the abundance of the miRNA signature set, haemolysis level, and sex (Appendix A), and saw no difference in correlation for male vs. female individuals. To address any potential bias from using all reproductively aged volunteers, we also ensured all miRNAs included here had previously been identified in multiple tissue types and were not affected by developmental stage.

Interestingly, all signature miRNAs, with the exception of miR-325-5p, have previously been reported as prognostically valuable plasma or serum biomarkers. In this small sampling of recent miRNA-biomarker research, we identified several instances where more than one of our haemolysis signature miRNAs were identified as disease biomarkers for the same condition in the same experiment [44,45,46], which, given our findings and those of previous haemolysis research, further call into question their validity as biomarkers of disease or condition. In conjunction with our research, we found many miRNAs as suggested circulating biomarkers for multiple disease states. For example, miR-122-5p was given biomarker potential in liver disease, lung cancer, and myasthenia gravis [14,44,47], and miR-660-5p was given biomarker potential in Alzheimer’s disease, breast cancer, and lung cancer [45,48,49], respectively. These miRNAs may represent effective biomarkers, but they may simply highlight RBC contamination or be indicative of a general state of inflammation.

This study has limitations. Firstly, data contained in this study were obtained from two cohorts (one pregnant, the other non-pregnant) of female volunteers of reproductive age. We attempted to mitigate this by validating our approach using two publicly available human plasma miRNA datasets: one male and one mixed male and female. Further, we have generalized this method such that removal (from the signature miRNA set) of domain-specific miRNA is built in, providing a framework that allows use within research conducted in any human plasma context. Secondly, the plasma preparation protocol, the centrifugation step, differs between the pregnant and non-pregnant groups, which confounds the interpretation of differentially abundant miRNAs associated with pregnancy. Our results highlight that ignoring the issue of miRNA from RBCs leaves researchers open to the risk that newly discovered miRNA disease biomarkers could in fact be biomarkers of haemolysis. Future research, including validation of the miRNA signature set proposed here using RT-qPCR, would strengthen confidence in our approach. Our research both recommends and enables tests for haemolysis to become standard pre-analytical practice in the absence of a physical assay for RBC contamination.

## Figures and Tables

**Figure 1 genes-13-01288-f001:**
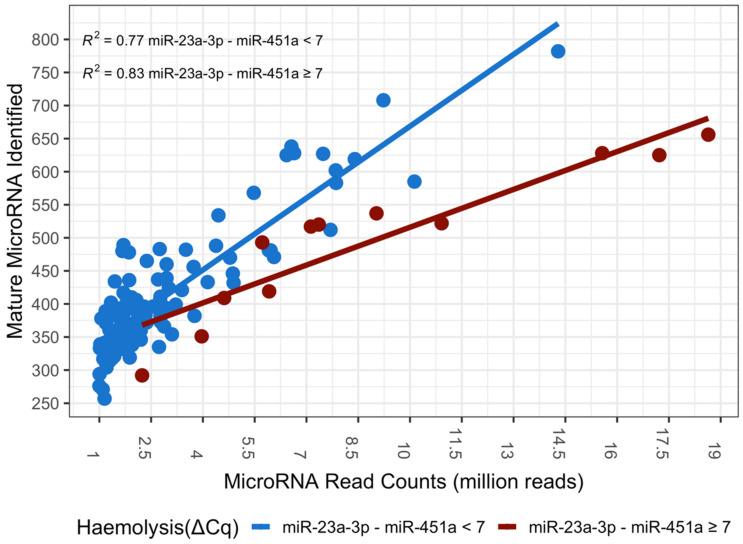
The number of mature miRNA species identified in an individual sample increases with read depth for both haemolysed (dark red) and non-haemolysed (blue) samples. However, the number of mature miRNA species identified for a given read depth is significantly lower (ANOVA, *p*-value = 1.68 × 10-9) in samples affected by haemolysis when compared to a non-haemolysed sample of equal read depth (*n* = 121).

**Figure 2 genes-13-01288-f002:**
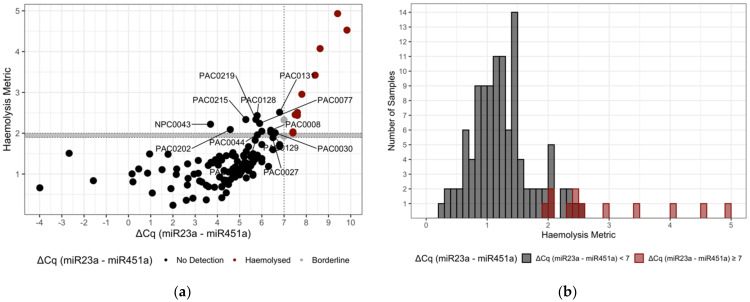
(**a**) A comparison of the derived haemolysis metric and the ΔCq measure of haemolysis shows a clear correlation. We identified 13 samples (named) that we suggest should be discarded or used with caution in further analysis. (**b**) Histogram of haemolysis metric values from the 121 samples in our experiment, coloured according to their ΔCq (miR-23a-3p-miR-451a) classification, indicate a minimum haemolysis metric of ≥ 1.9 for samples previously identified as haemolysed.

**Table 1 genes-13-01288-t001:** Patient Characteristics. Age, BMI and Gestational Age presented as mean and interquartile range.

	Pregnant		Not Pregnant	
	(*n* = 111)		(*n* = 10)	
Smoker	Yes	No	Yes	No
	48	63	unknown	unknown
Age	26.9 (21–32)		24.1 (21–24)	
BMI	24.3 (21–36)		23.8 (19–25)	
Gestational age (weeks)	12.5 (9–16)		Not applicable	

BMI = body mass index.

**Table 2 genes-13-01288-t002:** Twenty miRNAs with a general-use plasma haemolysis signature set. To remove confounding effects within our pregnancy-specific dataset, we identified a subset of 10 abundant miRNAs that were invariant with respect to pregnancy.

miRNA	Log_2_FC	Average Expression (log_2_ CPM)	Adjusted *p*-Value	Pregnancy Assoc.
miR-106b-3p	1.589	8.731	8.61 × 10^−15^	no
miR-140-3p	1.073	10.098	2.75 × 10^−13^	no
miR-142-5p	0.962	10.651	4.96 × 10^−12^	no
miR-532-5p	1.288	7.237	4.96 × 10^−12^	no
miR-17-5p	0.952	7.892	7.84 × 10^−12^	no
miR-19b-3p	1.128	8.696	1.93 × 10^−09^	no
miR-30c-5p	0.950	7.325	2.48 × 10^−09^	no
miR-324-5p	1.304	7.186	2.50 × 10^−09^	no
miR-192-5p	0.941	8.944	1.37 × 10^−08^	no
miR-660-5p	1.305	7.620	3.45 × 10^−10^	no
miR-186-5p	1.228	8.052	2.75 × 10^−13^	yes
miR-425-5p	1.282	11.246	4.96 × 10^−12^	yes
miR-25-3p	1.212	12.939	1.26 × 10^−11^	yes
miR-363-3p	1.237	7.882	4.52 × 10^−11^	yes
miR-183-5p	1.550	9.382	9.34 × 10^−11^	yes
miR-451a	1.372	13.002	3.65 × 10^−10^	yes
miR-182-5p	1.341	10.585	2.48 × 10^−09^	yes
miR-191-5p	0.929	11.790	4.68 × 10^−09^	yes
miR-194-5p	0.937	7.679	1.85 × 10^−08^	yes
miR-20b-5p	0.932	7.430	1.96 × 10^−08^	yes

## Data Availability

The dataset(s) supporting the conclusions of this article are available in NCBI’s Short Read Archive (SRA) (36) and are available through BioProject accession numbers PRJNA635621 and PRJNA824637.

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
