# Peer review of "Haemolysis Detection in MicroRNA-Seq from Clinical Plasma Samples"

_genes, 2022, doi:10.3390/genes13071288_

Round 1

Reviewer 1 Report

The authors of this study evaluated hemolysis detection in microRNA-seq samples

While the results are potentially interesting, they are mainly limited by the small sample size of the study including a great differences of 5 times, in the size of pregnant condition group.

As a potential consequence, the proportion of patients non-pregnant /pregnant is extremely different, which makes the results of this study difficult to be generalized.

Authors are recommended for the presentation of their works, however there are some important flaws that limit the potential publication. In particular, I guess that the design is not appropriate. I would suggest to include different validation cohort (including male population) and to perform Droplet Digital PCR in an independent cohort of patients to confirm miRNA signature discovery panel.  In the current state, the present author cannot recommend publication in Genes.

I do have the following general comments:

Methods

It is not clear how patients were selected for study enrolment and  the inclusion and exclusion criteria; (A paragraph that describes population information and mentioned age,ethnicity, familiarity, etc..is missing)

Results/discussion

The model is not tested against another independent cohort of patients. Moreover, to confirm the data model, it would be useful to compare the differentially expressed miRNA in a large cohort composed of male individuals.

Author Response

Response to comments by Reviewer #1

We thank the reviewer for the insightful and useful comments which have led us to substantially improve the manuscript including the addition of several new supplementary analyses. We have addressed the points raised individually below.

  1. While the results are potentially interesting, they are mainly limited by the small sample size of the study including a great differences of 5 times, in the size of pregnant condition group. As a potential consequence, the proportion of patients non-pregnant /pregnant is extremely different, which makes the results of this study difficult to be generalized.

The purpose of this study was to identify a robust set of microRNAs from small RNA-seq whose abundance was indicative of haemolysis in plasma samples. The evaluation of differential abundance in pregnant vs non-pregnant individuals was secondary to this and only for the purpose of illustrating that a researcher should identify and exclude from the signature any microRNAs whose differential abundance related to the biology of their dataset. For the purpose of identifying differentially abundant microRNAs in pregnancy, we believe our sample sizes were sufficient.

To further demonstrate that the microRNA signature set and Haemolysis metric can be generalised to identify haemolysis in other datasets, we have added supplementary analyses of two additional public datasets (for prostate cancer and arthritis; Supplementary Figure 6 & 7).

  1. I would suggest to include different validation cohort (including male population).

Thank you for this suggestion. We agree this was an important oversight and have provided two additional analyses (SupplementaryFigure 6-7) based on publicly available datasets for a) prostate cancer (all males) and b) early radiographic knee osteoarthritis (both male and female individuals). For each of these, as there was no haemolysis assessment available, we calculated a proxy measure comparable to the Cq measure by subtracting the log2 CPM expression of the invariant miRNA miR-23a-3p from the red blood cell associated miR-451a.

Each of these datasets show a strong concordance between the differential abundance of the signature microRNAs and haemolysis levels and importantly, there is not sex-specific differences in this relationship in the osteoarthritis dataset where both male and female individuals are present. 

As a side point, we noted that both datasets show evidence of differential abundance of miR-191-5p, one of the microRNAs from the signature set, relating to the disease in question. Interestingly, miR-191-5p has been shown to be differentially regulated in both prostate cancer and osteoarthritis (reference in the main paper). This highlights the need for a robust literature search prior to the haemolysis assessment using the Haemolysis metric proposed here. In these cases, we would recommend removing these microRNAs before calculating the Haemolysis metric, as we have done for our pregnancy-related microRNAs in our study.

  1. Perform Droplet Digital PCR in an independent cohort of patients to confirm miRNA signature discovery panel.

In our study, we evaluated the abundance of red blood cell-associated miR-451a using qPCR. While we agree that the proposed experiments would provide additional confidence in the microRNAs we have chosen, we do not have access to an additional cohort to perform this experiment.

Therefore, we have attempted to address this concern by including substantial additional validation of the generalisability and accuracy of the microRNA signature set and Haemolysis metric in the revised manuscript. Specifically, in addition to showing that the microRNA signature set generalises to two additional datasets (prostate cancer and osteoarthritis), we have added validation of the Haemolysis metric against a spectrophotometry absorbance method as well as the previous qPCR method (Supplementary Figure 5). We hope the reviewer will find that this new evidence provides an acceptable level of confidence in the generalisability and usefulness of the microRNA signature set and Haemolysis metric.

General comments

  1. Methods - It is not clear how patients were selected for study enrolment and the inclusion and exclusion criteria; (A paragraph that describes population information and mentioned age, ethnicity, familiarity, etc.is missing)

Thank you for bringing this to our attention, we have made the suggested changes in the Methods section, 2.1 Patient information.

  1. Results/discussion - The model is not tested against another independent cohort of patients. Moreover, to confirm the data model, it would be useful to compare the differentially expressed miRNA in a large cohort composed of male individuals. 

As described for Point #1 above, we agree and have included two new analyses in Supplementary Figures 6 and 7.

Reviewer 2 Report

This study is devoted to the assessment of haemolysis as a factor influencing plasma microRNA profiles. This important problem is related to the standardization of all studies of extracellular circulating miRNAs. At the moment, when studying miRNAs as biomarkers of various pathologies, researchers cannot accurately identify which changes are caused by the pathology itself and which are caused by technical factors, one of the key places among which is RBC haemolysis. Some published papers declare sets of haemolysis-related miRNAs such as miR-451a, miR-486-5p, miR-92a-3p, etc as circulating biomarkers of cancers, CVD, and other diseases and do not even include the word "h(a)emolysis" in the text.

Similar studies have been published before. The novelty of this study is that by using HTS on large sample size, the authors identified microRNAs associated with haemolysis and used this list of microRNAs for developing a bioinformatic tool to evaluate haemolysis in miRNA-seq of plasma samples.

However, a closer look at this study reveals many inconsistencies that should be explained, corrected, or separately pointed out in the text as limitations of the study and the applicability of its results.

Major comments

  1. Sample groups had different protocol for plasma isolation: double centrifugation at 800 x g for 15 minutes at 4°C for pregnancy termination samples and single centrifugation at 1015 x g for 10 minutes at 4°C for non-pregnant volunteers. The general sedimentation conditions differed more than 2-fold (24000 g*min vs. 10150 g*min). So, the differences in miRNA plasma profiles between these two groups could be not only due to biological but also due to pre-analytical factors.
  2. Only qPCR-based miR-451a/miR-23a-3p ratio was used for haemolysis assessment. The spectrophotometric measurement of free oxyhemoglobin at 414 nm wavelength was not performed for plasma samples in this study, however this method is quick and does not require any specific reagents, and is suitable to detect even low-level RBC haemolysis [1]. The absence of this step in this study leaves us unaware of how different techniques for measuring hemolysis relate in this sample and how this can be used in other samples.
  3. In SRA dataset samples, HiSeq 2500 is pointed as the sequencing method, however in the manuscript NextSeq is mentioned. Please clarify this issue. Also, it should be clarified whether all sample libraries were sequenced in one batch or there were different sequencing runs. If second, the number of runs with the number of samples in a single run should be mentioned. It should be mentioned if samples from pregnant and non-pregnant groups were sequenced in group batches in different runs.
  4. The general statistics of miRNA-Seq analysis in p. 3.1 lack the information about total miRNA read statistics. According to Table S1, the percentage of miRNA reads to total reads varied greatly from 11 to 97%, and about 20% of samples had less than one-third of miRNA reads. The information about the distribution of small RNA sequencing reads (miRNA, tRNA, mRNA, YRNA, and unaligned) in a graphical or table view should be presented in a manuscript for a better understanding of sequencing results.
  5. Why the authors considered samples with < 1 million reads to be unreliable due to low sequencing output? When analyzing miRNA-seq, one should look first at the total miRNA reads in the sample, so what did you mean by “low sequencing output”? After excluding 31 samples, about one-third of samples still had <1 mln miRNA reads (Table S1). Perhaps the right strategy would be to exclude samples with less than 1 mln miRNA reads, or with a percentage of miRNA reads less than some given value?
  6. It is unclear how the size of the groups changed during the exclusion of samples. The initial sample was 154 samples including 14 samples with ΔCq of 7 or above. After excluding 31 samples, 123 samples were to remain. However, PRJNA824637 set contain only 58 samples, and Figure 2b contain 121 samples. You need to specify exactly how many samples were left in each group (“haemolysed” and “non-haemolysed”). It is unclear what was the final sample size for differential expression analysis. In Fig. 1, please clarify the number of samples in each group.
  7. The authors do not discuss the possible reasons why some samples appeared to have increased haemolysis. Did these samples (“Haemolysis”, “Neutral” and “Caution” groups) have any distinct clinical or biological features, or did the haemolysis rate increase in vitro during sample preparation? In addition, here we are not talking about hemolysis per se, but about a shift in the qPCR-based ratio of the two microRNAs, which can change not only because of haemolysis. If haemolysed samples belonged to pregnant women, maybe the miR-451a/miR-23a shift in plasma was caused by some specific clinical parameters of pregnancy? This must be analyzed and discussed in the manuscript.
  8. The information on the difference in the dCt(miR-451a-miR-23a) value between pregnant and non-pregnant groups should be provided with statistical significance of comparison. What was the distribution of pregnant and non-pregnant samples in the group with ΔCq (miR-23a-miR-451a)>7?
  9. If 10 haemolysis-related miRNAs associated with pregnancy were excluded from the “signature” set, why they were not excluded also from the “background” set in the ‘Haemolysis Metric’ formula?
  10. The concordance between PCR and sequencing detection of the miR-451a/miR-23a ratio is unclear. This issue can be solved by the comparison of ΔCq (miR-23a-miR-451a) values and delta(log2CPM) values of these miRNAs, for example on a simple dot plot.
  11. The qPCR method of miRNA detection must be described in more detail (TaqMan of Sybr, the names of cDNA and qPCR kits, primer sequences, amplification steps, qPCR instrument, internal controls, etc).
  12. The discussion section lacks the comparison of obtained results with several important studies on hemolysis assessment in plasma miRNA profiles [1,2,3].

In conclusion, based on these comments, the limitations of the study should be clearly described in the manuscript. These limitations mean that the results of this study and the developed bioinformatic tool for haemolysis detection should be used with caution.

Minor comments

  1. Lines 210-213 in p. 3.1 should be moved to Materials and Methods section. The information of the sequencing platform used (NextSeq, 75 cycles) should be mentioned in p. 2.2
  2. In Table S1, there are some miRNAs that were not mentioned in the manuscript, namely hsa-miR-103a-3p, hsa-miR-191-5p, and hsa-miR-30c-5p. The detection of these miRNAs should be described in the manuscript, or the data on the Cq values of these miRNAs should be removed. The same situation with Uni-Sp6, UniSp-100, and UniSp-101.
  3. Please use the full name of miRNA (miR-23a-3p instead of miR-23a) in the manuscript text and figures.
  4. Were miRNA CPM values calculated per million total reads or per million miRNA reads?
  5. Please remove the duplicated column “Sample name” in Table S1
  6. ΔCq (miR-23a-miR-451a) values should be added to table s1 according to line 215
  7. It seems to me that it would be more informative if Figure 1 contained not Filtered read counts on the x-axis, but total miRNA read number, since this is what really determines the final diversity of microRNAs in the sample. Or this two plots can be represented both.
  8. Lines 425-426: the description of Table S1 lacks the information that sequencing statistics is presented in this table. Please add this to the Table S1 legend or separate Table S1 into two different tables with sequencing and qPCR data.
  9. The information on the percent of reads of miRNAs that are highly abundant in plasma (miR-451a and miR-486-5p) to the total amount of miRNA reads will be helpful

References

  1. Appierto V, Callari M, Cavadini E, Morelli D, Daidone MG, Tiberio P. A lipemia-independent NanoDrop®-based score to identify hemolysis in plasma and serum samples. Bioanalysis. 2014 May;6(9):1215-26.
  2. Landoni E, Miceli R, Callari M, Tiberio P, Appierto V, Angeloni V, Mariani L, Daidone MG. Proposal of supervised data analysis strategy of plasma miRNAs from hybridisation array data with an application to assess hemolysis-related deregulation. BMC bioinformatics. 2015 Dec;16(1):1-0.
  3. Shkurnikov MY, Knyazev EN, Fomicheva KA, Mikhailenko DS, Nyushko KM, Saribekyan EK, Samatov TR, Alekseev BY. Analysis of plasma microRNA associated with hemolysis. Bulletin of experimental biology and medicine. 2016 Apr;160(6):748-50.

Author Response

Response to comments by Reviewer #2

We thank the reviewer for the insightful and useful comments which have led us to substantially improve the manuscript including the addition of several new supplementary analyses. We have addressed the points raised individually below.

However, a closer look at this study reveals many inconsistencies that should be explained, corrected, or separately pointed out in the text as limitations of the study and the applicability of its results.

Major comments

  1. Sample groups had different protocol for plasma isolation... So, the differences in miRNA plasma profiles between these two groups could be not only due to biological but also due to pre-analytical factors.

You are correct in that pregnancy is confounded with spin speed, this was an unintended consequence of the length of the study. However, by removing miRNAs that were DE between pregnant and non-pregnant individuals we expect that this should effectively exclude any microRNAs that may be differentially present due to the technical factor of spin speed. This does leave open the possibility that microRNA(s) which are excluded may not be differentially expressed in pregnancy however this is not the aim of this study and we believe this approach should not compromise the accuracy of the Haemolysis metric calculated from the remaining microRNAs.

  1. Only qPCR-based miR-451a/miR-23a-3p ratio was used for haemolysis assessment. The spectrophotometric measurement of free oxyhemoglobin at 414 nm wavelength was not performed for plasma samples in this study... The absence of this step in this study leaves us unaware of how different techniques for measuring hemolysis relate in this sample and how this can be used in other samples.

Thank you for the suggestion. We agree that the inclusion of spectrophotometry-based validation would greatly improve the interpretability study and have included additional validation experiments comparing the spectrophotometry absorbance (Log10(A414)) readings with the results of the Haemolysis metric (and qRT-PCR) methods (Supplementary Figure 5). Although this analysis is performed on a subset of samples (48 samples; due to the unavailability of the remaining samples), the number of samples used is sufficient to show high correlations between all three methods (R > 0.82). Interestingly, these analyses showed that Haemolysis metric is better correlated with the spectrophotometry method (R = 0.87) than the spectrophotometry method is with the gold-standard qRT-PCR method (R = 0.82), which is consistent the Haemolysis metric being an accurate and useful marker of haemolysis.

  1. In SRA dataset samples, HiSeq 2500 is pointed as the sequencing method, however in the manuscript NextSeq is mentioned. Please clarify this issue. Also, it should be clarified whether all sample libraries were sequenced in one batch or there were different sequencing runs. If second, the number of runs with the number of samples in a single run should be mentioned. It should be mentioned if samples from pregnant and non-pregnant groups were sequenced in group batches in different runs.

Thank you for bringing these oversights to our attention. The instrument used to sequence the new data not yet publicly available was the NovaSeq. Data previously sequenced and published by us (GSE151362) was run on the NextSeq. We have updated the manuscript to make this clear, and submitted a revision to the Short Read Archive to amend the record. We have also made the suggested changes in the Methods section of the main document and included information on sequencing batch in Supplementary Table 2.

  1. The general statistics of miRNA-Seq analysis in p. 3.1 lack the information about total miRNA read statistics... The information about the distribution of small RNA sequencing reads (miRNA, tRNA, mRNA, YRNA, and unaligned) in a graphical or table view should be presented in a manuscript for a better understanding of sequencing results.

We have included a table describing the distribution of small RNA sequencing reads across various RNA classes as gleaned from the GRCh38 Ensembl gene annotation.

  1. Why the authors considered samples with < 1 million reads to be unreliable due to low sequencing output? When analyzing miRNA-seq, one should look first at the total miRNA reads in the sample, so what did you mean by “low sequencing output”? After excluding 31 samples, about one-third of samples still had <1 mln miRNA reads (Table S1). Perhaps the right strategy would be to exclude samples with less than 1 mln miRNA reads, or with a percentage of miRNA reads less than some given value? ... It is unclear how the size of the groups changed during the exclusion of samples... You need to specify exactly how many samples were left in each group (“haemolysed” and “non-haemolysed”). It is unclear what was the final sample size for differential expression analysis. In Fig. 1, please clarify the number of samples in each group.

We thank the reviewer for identifying these shortfalls in our communication. For clarity, we have now excluded from Supplementary Table 1, 2 and 3 all samples which did not pass our quality control criteria and rephrased the manuscript to reflect this. Briefly, our experiment included counts from 121 miRNA-sequencing libraries prepared from human plasma with a total miRNA read count of > 1 million reads per sample. Of these data, 12 samples were identified through dCq (miR-451a - miR-23a-3p ≥ 7) as being haemolysed. We have also specified the number of samples being analysed in the legend of Figure 1.

  1. The authors do not discuss the possible reasons why some samples appeared to have increased haemolysis. Did these samples (“Haemolysis”, “Neutral” and “Caution” groups) have any distinct clinical or biological features, or did the haemolysis rate increase in vitro during sample preparation? In addition, here we are not talking about hemolysis per se, but about a shift in the qPCR-based ratio of the two microRNAs, which can change not only because of haemolysis. If haemolysed samples belonged to pregnant women, maybe the miR-451a/miR-23a shift in plasma was caused by some specific clinical parameters of pregnancy? This must be analyzed and discussed in the manuscript.

We have presented the limited maternal characteristics information we have available in Table 1, including age, BMI, smoking status and gestational age. We performed students’ t-tests on dCq (miR-451a - miR-23a-3p) both within the Pregnant and Not Pregnant groups and across the cohort as a whole  for each characteristic, splitting Age, BMI and Gestational Age into equal-sized groups.   We observed no significant association (all p-values > 0.29) with any of these characteristics which might explain the variation in dCq values or correlated changes in miRNA abundance. We do, however, acknowledge that these are a limited set of characteristics and this does not preclude the possibility some fraction of the microRNA abundance changes relate to an unmeasured maternal characteristic.

  1. The information on the difference in the dCt(miR-451a-miR-23a) value between pregnant and non-pregnant groups should be provided with statistical significance of comparison. What was the distribution of pregnant and non-pregnant samples in the group with ΔCq (miR-23a-miR-451a)>7?

We appreciate the reviewer’s perspective but find that answering this question is complex and respectfully disagree that the requested information is important to our study. First, we clarify that there are substantial differences between dCt (miR-23a-miR-451a) distributions for the Pregnant and Not Pregnant groups, including that none of the Not Pregnant samples have dCt(miR-23a-miR-451a) > 7. However, as part of the study, we determined that miR-451a is statistically significantly differentially abundant between pregnant and non-pregnant individuals (Table 2) which may bias the requested statistical evaluation of a dCt(miR-23a-miR-451a).

Fortunately however, we do not feel that this interferes with the purpose of our study; which was to identify microRNAs which can be used as a measure of Haemolysis and to intentionally avoid using microRNAs that are differentially abundant between our conditions (pregnancy). In light of the above, we respectfully suggest it will not strengthen our study to perform statistical evaluation of the dCt(miR-23a-miR-451a) values between the Pregnant and Non Pregnant groups.

  1. If 10 haemolysis-related miRNAs associated with pregnancy were excluded from the “signature” set, why they were not excluded also from the “background” set in the ‘Haemolysis Metric’ formula?

The point the reviewer has made is correct, in fact our method does recommend excluding biology-associated microRNAs (in this case, pregnancy associated) from the background set as well as the signature set. We appreciate that the reviewer's comments have drawn to our attention that this was unclear and have clarified this in the section defining the Haemolysis Metric.

  1. The concordance between PCR and sequencing detection of the miR-451a/miR-23a ratio is unclear. This issue can be solved by the comparison of ΔCq (miR-23a-miR-451a) values and delta(log2CPM) values of these miRNAs, for example on a simple dot plot.

Thank you for the suggestion. We agree and have included a scatterplot of the Δlog2CPM (miR-451a-miR-23a-3p) as a function of the ΔCq (miR-451a-miR-23a-3p) in Supplementary Figure 4.3 clearly identifying a strong correlation between these two measure of haemolysis.

  1. The qPCR method of miRNA detection must be described in more detail (TaqMan of Sybr, the names of cDNA and qPCR kits, primer sequences, amplification steps, qPCR instrument, internal controls, etc).

Details of the qPCR materials, methods and instruments have been added to section 2.4.

  1. The discussion section lacks the comparison of obtained results with several important studies on hemolysis assessment in plasma miRNA profiles [1,2,3].

Thank you for bringing these to our attention, all three were highly relevant and we have added them to the introduction and discussion sections.

  1. The limitations of the study should be clearly described in the manuscript.

These limitations mean that the results of this study and the developed bioinformatic tool for haemolysis detection should be used with caution.

Minor comments

Thank you for these suggestions and observations, the manuscript has been updated to address these changes. For point #9, The CPM (normalised to total miRNA reads) of the 189 miRs analysed are presented as Supplementary Table 4; this includes miR-451a and miR-486-5p.

  1. Lines 210-213 in p. 3.1 should be moved to Materials and Methods section. The information of the sequencing platform used (NextSeq, 75 cycles) should be mentioned in p. 2.2
  2. In Table S1, there are some miRNAs that were not mentioned in the manuscript, namely hsa-miR-103a-3p, hsa-miR-191-5p, and hsa-miR-30c-5p. The detection of these miRNAs should be described in the manuscript, or the data on the Cq values of these miRNAs should be removed. The same situation with Uni-Sp6, UniSp-100, and UniSp-101.
  3. Please use the full name of miRNA (miR-23a-3p instead of miR-23a) in the manuscript text and figures.
  4. Were miRNA CPM values calculated per million total reads or per million miRNA reads?
  5. Please remove the duplicated column “Sample name” in Table S1
  6. ΔCq (miR-23a-miR-451a) values should be added to table s1 according to line 215
  7. It seems to me that it would be more informative if Figure 1 contained not Filtered read counts on the x-axis, but total miRNA read number, since this is what really determines the final diversity of microRNAs in the sample. Or this two plots can be represented both.
  8. Lines 425-426: the description of Table S1 lacks the information that sequencing statistics is presented in this table. Please add this to the Table S1 legend or separate Table S1 into two different tables with sequencing and qPCR data.
  9. The information on the percent of reads of miRNAs that are highly abundant in plasma (miR-451a and miR-486-5p) to the total amount of miRNA reads will be helpful

Reviewer 3 Report

Smith and colleagues have sought to investigate whether a miRNA haemolysis signature can be identified from RNA-seq data collected from plasma as a way of assessing whether there was significant haemolysis occurred in the samples. This is an important consideration as red blood cells can contain contaminate samples with miRNA and overwhelm the miRNA signal within the sample.

My experience in this field is as a researcher investigating small ncRNA expression, including miRNA, in serum for potential biomarkers. I feel that this work is invaluable, especially for using pre-existing datasets to determine if any of the samples are affected by potential haemolysis. However, I feel that it needs to be made clear that this approach is not suitable for new studies, as it seems a time-consuming and expensive approach to be investigating whether the samples have haemolysis after all the work, or as an additional measure to consider. An option that they have not mentioned is the use of ELISA’s to measure the level of haemolysis, something which I have used in my own studies. I feel this should be mentioned at the very least as it is a valid approach to do.

Related to this, a major issue is that both the RT-qPCR which they used to define haemolysed samples and the RNA-seq analysis is that these are in some ways indirect measures of haemolysis. Considering they note that the miR-451a was affected by pregnancy, there is a potential this may affect their definition of a haemolysed sample. As such, in my opinion, the only way they can reach a valid conclusion to say that these miRNA are related to haemolysis is to show through another method that does not involve RNA, otherwise I feel that there is a level of self-fulfilment.

I think that it is especially important for a study like this that this claiming to have a signature for haemolysis that at the very least there are correlative analyses done with a more direct measure of this, either through the use of spectrophotometer or an ELISA against haemoglobin, the latter using 2-5ul of samples. I would recommended that either, but ideally both, are done and compared to both the RNA-seq and RT-qPCR data as that would provide more clarity about the thresholds and provide multiple points of reference. Without this, I feel the study has reduced value.

The other question that I had was whether the changes in these miRNA between haemolysed and non-haemolysed samples confirmed using RT-qPCR? It is important to validate that the differences observed are confirmed and the results match up with that from the RNA-seq.

I recommend that these changes are made as they are critical to build confidence in the data presented in this paper. As such, I recommend this manuscript to be revised and resubmitted.

Author Response

Reviewer 3

We thank the reviewer for the insightful and useful comments which have led us to substantially improve the manuscript including the addition of several new supplementary analyses. We have addressed the points raised individually below.

Comments and Suggestions for Authors

Smith and colleagues have sought to investigate whether a miRNA haemolysis signature can be identified from RNA-seq data collected from plasma as a way of assessing whether there was significant haemolysis occurred in the samples. This is an important consideration as red blood cells can contain contaminate samples with miRNA and overwhelm the miRNA signal within the sample.

My experience in this field is as a researcher investigating small ncRNA expression, including miRNA, in serum for potential biomarkers. I feel that this work is invaluable, especially for using pre-existing datasets to determine if any of the samples are affected by potential haemolysis. However, I feel that

  1. It needs to be made clear that this approach is not suitable for new studies, as it seems a time-consuming and expensive approach to be investigating whether the samples have haemolysis after all the work, or as an additional measure to consider.

You are correct, we do not expect that this will replace traditional methods of analysing haemolysis as a due diligence step prior to sequencing analysis and have amended the manuscript to make the usage clearer.

Our intention is that researchers will use this method both for post-hoc analyses of haemolysis levels on datasets as a second confirmation of a previous result; or for which this was not previously not considered or reported; or where the sequencing will go ahead regardless of haemolysis (sample is too valuable); or where, for example, it is later suspected that the original analyses was compromised (for example, the contamination of spectrophotometry samples or biology-driven differential abundance the qPCR gold standard miR-451a, as was the case for pregnancy). In cases such as these, we believe our method can be very valuable.

  1. An option that they have not mentioned is the use of ELISA’s to measure the level of haemolysis, something which I have used in my own studies. I feel this should be mentioned at the very least as it is a valid approach to do. 

Thank you for drawing this to our attention, we have amended the text accordingly.

  1. Related to this, a major issue is that both the RT-qPCR which they used to define haemolysed samples and the RNA-seq analysis is that these are in some ways indirect measures of haemolysis. Considering they note that the miR-451a was affected by pregnancy, there is a potential this may affect their definition of a haemolysed sample. As such, in my opinion, the only way they can reach a valid conclusion to say that these miRNA are related to haemolysis is to show through another method that does not involve RNA, otherwise I feel that there is a level of self-fulfilment.

We agree that this is a valid concern. To address this, we have included additional validation experiments (Supplementary Figure 5) comparing the spectrophotometry absorbance (Log10(A414)) readings with the results of the Haemolysis metric (and qRT-PCR) methods. Although this analysis is performed on a subset of samples (48 samples; this is because the assay was performed some time ago and we are unable to repeat it as we no longer have access to the samples), the number of samples is sufficient to show high correlations between all three methods (R > 0.82). Interestingly, these analyses showed that Haemolysis metric is better correlated with the spectrophotometry method (R = 0.87) than the spectrophotometry method is with the gold-standard qRT-PCR method (R = 0.82), which is consistent the Haemolysis metric being an accurate and useful marker of haemolysis.

  1. I think that it is especially important for a study like this that this claiming to have a signature for haemolysis that at the very least there are correlative analyses done with a more direct measure of this, either through the use of spectrophotometer or an ELISA against haemoglobin, the latter using 2-5ul of samples. I would recommended that either, but ideally both, are done and compared to both the RNA-seq and RT-qPCR data as that would provide more clarity about the thresholds and provide multiple points of reference.

Thank you for the suggestion. We agree that the inclusion of spectrophotometry-based validation would greatly improve the interpretability study and have included additional validation experiments as described above.

  1. The other question that I had was whether the changes in these miRNA between haemolysed and non-haemolysed samples confirmed using RT-qPCR? It is important to validate that the differences observed are confirmed and the results match up with that from the RNA-seq.

Our study, used RT-qPCR to evaluate the abundance of miR-23a-3p and miR451a (which is included in the miRNA signature set) and showed good general agreement between the dCt metric calculated from the RT-qPCR expression levels for these miRNAs and a proxy measure calculated from the expression levels of the same miRNAs from sequencing data (Supplementary Table 4.3). While we agree that the evaluation of the other 19 miRNAs in the haemolysis signature would provide additional confidence in the miRNAs we have chosen, the samples used in this study are no longer available so we are unable to perform this. We believe however, that the revised manuscript provides data for a sufficient level of confidence, given the following points. Firstly, that previous studies have shown that there is good general agreement between qPCR and sequencing technologies and that a strength of our approach is in the use of several miRNAs which guards against the end result being strongly biased by platform differences for individual miRNAs. Furthermore, the qPCR validation using the gold-standard miRNAs (miR-23a-3p and miR451a) and correlation of our Haemolysis metric with these results and with the second well regarded spectrophotometry absorbance method (see Supplementary Figure 5). We believe that this evidence together provides an acceptable level of confidence in the usefulness of the Haemolysis metric.

Round 2

Reviewer 1 Report

I thank the authors for their consideration of my comments, and for addressing many of my concerns

Author Response

The authors wish to thank the reviewer for taking the time to read our revised manuscript and note that there were no comments to be addressed in the second revision.

Reviewer 2 Report

The authors have done a great deal of work to improve the manuscript in accordance with previous comments. Most of the comments were taken into account to correct the text or add new data. However, some issues remain unresolved, unreported in the article, or insufficiently disclosed.

1.      «However, by removing miRNAs that were DE between pregnant and non-pregnant individuals we expect that this should effectively exclude any microRNAs that may be differentially present due to the technical factor of spin speed. This does leave open the possibility that microRNA(s) which are excluded may not be differentially expressed in pregnancy however this is not the aim of this study and we believe this approach should not compromise the accuracy of the Haemolysis metric calculated from the remaining microRNAs.»

You state that none of the Not Pregnant samples had dCt(miR-23a-miR-451a) > 7. This means that you already have bias toward one of the groups in the representation of samples with hemolysis. So, comparison between the pregnant and non-pregnant groups is very likely to result in some of the hemolysis-related microRNAs (and primarily the most highly represented ones) being differentially expressed between these groups. This is exactly what is further found in your data: the four most abundant DE miRNAs with the mean log2CPM > 11 (miR-425-5p, miR-25-3p, miR-451a, miR-191-5p) fall into the group with differential expression between the pregnant and non-pregnant groups. The presence of these miRNAs within the hemolysis associated set seems logical, as three of them are abundant in erythrocytes. It would make sense to compare the non-pregnant group with the pregnant subgroup of samples with deltaCt(miR-23a-3p-miR-451a) < 7 to obtain a set of miRNAs associated with pregnancy. Alternatively, the non-pregnant group from could be excluded from the study, since it is firstly too small compared to the pregnant group, and secondly has a different plasma preparation protocol. In addition, the subgroup "miRNAs associated with pregnancy" has no right to be called that in the text of the manuscript, because your groups of pregnant and non-pregnant women differed in both centrifugation conditions and the presence of samples with hemolysis. Please use the terms “DE miRNAs between the study groups” or “DE miRNAs between the NP and PAC groups” or some similar terms instead. In order to detect pregnancy-associated microRNAs, the authors initially had to use two samples of similar size, with as similar age as possible, with the same plasma obtaining conditions, excluding all samples with hemolysis, and preferably sequencing the samples in the same batch on the same instrument.

2.      The limitations of the study should be clearly described in the “Discussion” section, using the term “Limitations of the study” in the text. Without this, the study cannot be recommended for publication. Please state this in the text according to the major comments 1 and 7 from the first review and the comment #1 above.

3.      There are inconsistencies in Table S2, namely, Batch 3 (42 samples) contains 10 NP samples made on NovaSeq and 32 PAC samples made on NextSeq 500. In the text, however, you indicate that all 42 new samples were sequenced on NovaSeq. Please clarify and correct this.

4.      SRA data still mention Illumina HiSeq 2500 in the description, however, as you stated, NovaSeq system was used for sequencing

5.      The response to the comment â„–5: “For clarity, we have now excluded from Supplementary Table 1, 2 and 3 all samples which did not pass our quality control criteria and rephrased the manuscript to reflect this.”

The quality control criteria are not described clearly in the manuscript.

Lines 275-277: “An average of ~2.9 million reads were sequenced per sample (range ~0.25-18.6 million reads) and we selected libraries with < 1 million reads for further analyses.” This must be corrected. In your response you stated that you selected samples with > 1 mln miRNA reads. Please clarify this issue.

6.      In the study you postulate that you developed the value “Haemolysis Metric” that can be calculated based on plasma miRNA-seq data. Why don't you use this particular value when validating on external datasets? Instead, you provide 20 plots for each individual miRNA from your DE set. I recommend to use Haemolysis Metric value for external dataset validation. It can be showed in the single plot for each dataset. You postulate that “recommend excluding from both the signature set and background any miRNAs known or suspected to relate to the study biology before calculating the Haemolysis Metric”. As you use datasets with the described biological features, you should demonstrate on these datasets how this exclusion may work in real datasets by exclusion miRs linked with PCa in the GSE118038 set, and linked with osteoarthritis in the GSE151341 set.

7.      Validation of the methodology you suggested in the article for evaluating hemolysis on external datasets is highly appreciated. However, visualization of the results in the Supplementary Figures 4.1, 4.2, 4.3, 6 and 7 lack legends as well as numbers and captions on the x- and y-axes. Please indicate in the text description what corresponds to the figures beginning with “a” and ending with “t”. Please correct the dataset number from GSE11803 to GSE118038 in the manuscript text (line 554) and in Figure 7. Please indicate the sample size in Figures 6 and 7.

8.      Lines 274-275: “We identified 14 plasma samples with a ΔCq of 7 or above (Supplementary Table 2).” However, Supplementary Table 2 contains only 12 samples with a ΔCq of 7 or above. As I mentioned in my first review, you should clearly state how the initial sample size (121 samples) changed after using the filter for samples with the miRNA read count > 1 mln.

9.      Lines 129-130: the sentence should be corrected as it contains repeated word “women”

10.   Table 2: please use the same number of decimal places in all numbers.

Author Response

Response to comments by Reviewer #2

The authors have done a great deal of work to improve the manuscript in accordance with previous comments. Most of the comments were taken into account to correct the text or add new data. However, some issues remain unresolved, unreported in the article, or insufficiently disclosed.

  1.     «However, by removing miRNAs that were DE between pregnant and non-pregnant individuals we expect that this should effectively exclude any microRNAs that may be differentially present due to the technical factor of spin speed. This does leave open the possibility that microRNA(s) which are excluded may not be differentially expressed in pregnancy however this is not the aim of this study and we believe this approach should not compromise the accuracy of the Haemolysis metric calculated from the remaining microRNAs.»

You state that none of the Not Pregnant samples had dCt(miR-23a-miR-451a) > 7. This means that you already have bias toward one of the groups in the representation of samples with hemolysis. So, comparison between the pregnant and non-pregnant groups is very likely to result in some of the hemolysis-related microRNAs (and primarily the most highly represented ones) being differentially expressed between these groups. This is exactly what is further found in your data: the four most abundant DE miRNAs with the mean log2CPM > 11 (miR-425-5p, miR-25-3p, miR-451a, miR-191-5p) fall into the group with differential expression between the pregnant and non-pregnant groups. The presence of these miRNAs within the hemolysis associated set seems logical, as three of them are abundant in erythrocytes. It would make sense to compare the non-pregnant group with the pregnant subgroup of samples with deltaCt(miR-23a-3p-miR-451a) < 7 to obtain a set of miRNAs associated with pregnancy. Alternatively, the non-pregnant group from could be excluded from the study, since it is firstly too small compared to the pregnant group, and secondly has a different plasma preparation protocol. In addition, the subgroup "miRNAs associated with pregnancy" has no right to be called that in the text of the manuscript, because your groups of pregnant and non-pregnant women differed in both centrifugation conditions and the presence of samples with hemolysis. Please use the terms “DE miRNAs between the study groups” or “DE miRNAs between the NP and PAC groups” or some similar terms instead. In order to detect pregnancy-associated microRNAs, the authors initially had to use two samples of similar size, with as similar age as possible, with the same plasma obtaining conditions, excluding all samples with hemolysis, and preferably sequencing the samples in the same batch on the same instrument.

Thank you for your comments. When comparing the pregnant and non-pregnant groups for differential miRNA abundance we did indeed only compare samples with deltaCt(miR-23a-3p-miR-451a) < 7 such that haemolysis and pregnancy were not confounded, and have adjusted the manuscript to explicitly state this (lines 229-230 of the revised manuscript). Regarding the question of miRNA abundance differences between our PAC (pregnant) and NPC (non-pregnant) cohorts, we have clarified the differential abundance used in our analyses to include the explicit use of the “PAC” and “NPC” cohort names (revised manuscript lines 340:341).

Regarding the different number of sample sizes, our statistical approach (limma) does not require equal sample sizes (as stated by Aaron Lun (WEHI, Australia), one of the authors of the R package in the following support post: https://support.bioconductor.org/p/69345/). It is also designed to have good statistical power with small sample sizes (Section 8.1, pg 36-37, Limma User’s Guide from the vignette on Bioconductor).

  1.      The limitations of the study should be clearly described in the “Discussion” section, using the term “Limitations of the study” in the text. Without this, the study cannot be recommended for publication. Please state this in the text according to the major comments 1 and 7 from the first review and the comment #1 above.

An additional statement regarding the suggested study limitations and our efforts to mitigate said limitations has been added to the manuscript discussion as suggested (revised manuscript lines 566-759).

  1.     There are inconsistencies in Table S2, namely, Batch 3 (42 samples) contains 10 NP samples made on NovaSeq and 32 PAC samples made on NextSeq 500. In the text, however, you indicate that all 42 new samples were sequenced on NovaSeq. Please clarify and correct this.

As you note, this was an error and has now been corrected. SupplementaryTable2.xls will be resubmitted.

  1.     SRA data still mention Illumina HiSeq 2500 in the description, however, as you stated, NovaSeq system was used for sequencing

The authors apologize that this was still unclear in the manuscript. The samples from our lab are from two datasets, both of which are held as part of the NCBI database. Sample data can be found under BioProject numbers PRJNA635621 (previously published by the authors) and PRJNA824637. This additional information, along with the individual sample Accession numbers has been added to Supplementary Table 2 and further detail has been added to the Data availability statement (revised manuscript lines 276 and 655-657).

  1.     The response to the comment â„–5: “For clarity, we have now excluded from Supplementary Table 1, 2 and 3 all samples which did not pass our quality control criteria and rephrased the manuscript to reflect this.” 

The quality control criteria are not described clearly in the manuscript. 

Lines 281-286: “An average of ~2.9 million reads were sequenced per sample (range ~0.25-18.6 million reads) and we selected libraries with < 1 million reads for further analyses.” This must be corrected. In your response you stated that you selected samples with > 1 m miRNA reads. Please clarify this issue.

Thanks for your observations, this section has been clarified as follows (revised manuscript lines 281-286): “Using libraries with > 1 million reads for analyses, we obtained 121 libraries with an average of ~3.49 million reads per sample (range ~1.00-18.64 million reads).“

  1.     In the study you postulate that you developed the value “Haemolysis Metric” that can be calculated based on plasma miRNA-seq data. Why don't you use this particular value when validating on external datasets? Instead, you provide 20 plots for each individual miRNA from your DE set. I recommend to use Haemolysis Metric value for external dataset validation. It can be showed in the single plot for each dataset. You postulate that “recommend excluding from both the signature set and background any miRNAs known or suspected to relate to the study biology before calculating the Haemolysis Metric”. As you use datasets with the described biological features, you should demonstrate on these datasets how this exclusion may work in real datasets by exclusion miRs linked with PCa in the GSE118038 set, and linked with osteoarthritis in the GSE151341 set. 

Thank you for the suggestion, we agree that this would add value to the manuscript. We had calculated the Haemolysis metric as part of the validation studies (and coloured the plots by it) but had not explicitly plotted it. We have now included additional plots in Supp Fig 6 and 7 which show correlations for the miRNA signature set miRNAs that were excluded from the Haemolysis Metric calculations (Supp Fig 6.3 and 7.3) and the relationship between the proxy miR-23a-3p:miR-451a metric and the Haemolysis Metric (Supp Fig 6.2 and 7.2). We have adjusted the text in lines 481:441 of the revised manuscript to describe the additional figures.

  1.     Validation of the methodology you suggested in the article for evaluating hemolysis on external datasets is highly appreciated. However, visualization of the results in the Supplementary Figures 4.1, 4.2, 4.3, 6 and 7 lack legends as well as numbers and captions on the x- and y-axes. Please indicate in the text description what corresponds to the figures beginning with “a” and ending with “t”. Please correct the dataset number from GSE11803 to GSE118038 in the manuscript text (line 554) and in Figure 7. Please indicate the sample size in Figures 6 and 7.

The authors apologize for these missing details (numbers and captions on the x and y-axis) which were unintentionally lost during the conversion of the supplementary data to PDF format. This has been corrected by submission of the original word documents (new submission of files SuppFigure4.docx, SuppFigure6.docx, SuppFigure7.docs). Legends for all figures remain on the page trailing the figure to prevent the need to further reduce the size of the individual plots presented. Further information describing the figures (a-t) has been incorporated into the figure legends (included in the new submission of files SuppFig4.docx, SuppFig6.docx, SuppFig7.docs). The typographic error (GSE11803 to GSE118038) has been corrected (SuppFigure7.docx, line 623 of the revised manuscript). Sample size for Supplementary figures 6 and 7 have been added to the figure legend.

  1.     Lines 274-275: “We identified 14 plasma samples with a ΔCq of 7 or above (Supplementary Table 2).” However, Supplementary Table 2 contains only 12 samples with a ΔCq of 7 or above. As I mentioned in my first review, you should clearly state how the initial sample size (121 samples) changed after using the filter for samples with the miRNA read count > 1 mln. 

The authors apologize for this oversight. In the previous revision, we realised that the number of samples being used was confusing and so we adjusted our approach in an attempt to clarify the number of samples used. In doing so however, the number of haemolysed samples was changed from 14 to 12 and we failed to notice that we had not updated the text in the location you describe. We have adjusted it accordingly. The manuscript should now reflect that after filtering the samples, we retained 121 samples with > 1 million reads. Of these, 12 samples were found to be haemolysed.

  1.     Lines 129-130: the sentence should be corrected as it contains repeated word “women”

The authors apologize for the oversight. This has been corrected (line 134 of the revised manuscript).

  1.   Table 2: please use the same number of decimal places in all numbers.

The authors apologize for the oversight. The trailing zero in the third decimal place has been added to all 5 instances as required (Table 2 of the revised manuscript).

Reviewer 3 Report

I thank the authors for their consideration of my comments, and for addressing many of my concerns.

For final approval, I would request that the authors please in the text put the explanation they provided in response to my first question about how they envisage this analysis being used in the introduction - as I said previously, this needs to be made clear and evident to the reader and should only be undertaken when direct assessment of hemolysis cannot be done.

Additionally, I would ask the authors to please provide a comment in the discussion in relation to the lack of validation of the targets with RT-qPCR and how this should be carried out to confirm the RNA-seq data since RNA-seq data is not infallible.

If the authors can make these two changes, I would be happy to recommend acceptance of the paper.

Author Response

Response to comments by Reviewer #3

I thank the authors for their consideration of my comments, and for addressing many of my concerns.

For final approval, I would request that the authors please in the text put the explanation they provided in response to my first question about how they envisage this analysis being used in the introduction - as I said previously, this needs to be made clear and evident to the reader and should only be undertaken when direct assessment of hemolysis cannot be done.

Thank you for your comment. We have provided an additional statement in the introduction advising readers that our method provides the most value where the physical assay of haemolysis cannot be undertaken (line 122-124 of revised manuscript) and further clarified this in the discussion (line 591 of the revised manuscript). We note however, that whilst the most value is to be gained where access to the physical sample for assay is not possible/practical, we do believe that all relevant studies with existing miRNA-seq data would benefit from performing an in silico analysis of haemolysis using our method; there is no apparent downside - it is simple, free and provides an additional source of information to the researcher.

Additionally, I would ask the authors to please provide a comment in the discussion in relation to the lack of validation of the targets with RT-qPCR and how this should be carried out to confirm the RNA-seq data since RNA-seq data is not infallible.

Thank you for the suggestion. We have provided a statement regarding the issue of RT-qPCR validation in the concluding paragraph of the discussion (lines 577-589 of the revised manuscript).

If the authors can make these two changes, I would be happy to recommend acceptance of the paper.

Round 3

Reviewer 2 Report

The authors have corrected the manuscript in accordance with the comments. 

The results of this study should be used with caution, as it has a number of serious limitations outlined in the manuscript.

Minor comment:

- Supplementary figures 4.1, 4.2, 4.3, 6.1, 6.3, 7.1, and 7.3 lack captions for the X and Y axes, as well as units.